# GRAPH TRAVERSAL WITH TENSOR FUNCTIONALS: A META-ALGORITHM FOR SCALABLE LEARNING

**Elan Markowitz**[*,1,2], **Keshav Balasubramanian**[*,1], **Mehrnoosh Mirtaheri**[*,1,2], **Sami Abu-El-Haija**[*,1,2]
[*]Equal Contribution
[1]University of Southern California, [2]USC Information Sciences Institute

**Bryan Perozzi**
Google Research

**Greg Ver Steeg**[1,2], **Aram Galstyan**[1,2]

## ABSTRACT

Graph Representation Learning (GRL) methods have impacted fields from chemistry to social science. However, their algorithmic implementations are specialized to specific use-cases e.g. *message passing* methods are run differently from *node embedding* ones. Despite their apparent differences, all these methods utilize the graph structure, and therefore, their learning can be approximated with stochastic graph traversals. We propose Graph Traversal via Tensor Functionals (GTTF), a unifying meta-algorithm framework for easing the implementation of diverse graph algorithms and enabling transparent and efficient scaling to large graphs. GTTF is founded upon a data structure (stored as a sparse tensor) and a stochastic graph traversal algorithm (described using tensor operations). The algorithm is a functional that accept two functions, and can be specialized to obtain a variety of GRL models and objectives, simply by changing those two functions. We show for a wide class of methods, our algorithm learns in an unbiased fashion and, in expectation, approximates the learning as if the specialized implementations were run directly. With these capabilities, we scale otherwise non-scalable methods to set state-of-the-art on large graph datasets while being more efficient than existing GRL libraries – with only a handful of lines of code for each method specialization. GTTF and its various GRL implementations are on: `https://github.com/isi-usc-edu/gttf`

## 1 INTRODUCTION

Graph representation learning (GRL) has become an invaluable approach for a variety of tasks, such as node classification (e.g., in biological and citation networks; Veličković et al. (2018); Kipf & Welling (2017); Hamilton et al. (2017); Xu et al. (2018)), edge classification (e.g., link prediction for social and protein networks; Perozzi et al. (2014); Grover & Leskovec (2016)), entire graph classification (e.g., for chemistry and drug discovery Gilmer et al. (2017); Chen et al. (2018a)), etc.

In this work, we propose an algorithmic unification of various GRL methods that allows us to re-implement existing GRL methods and introduce new ones, in merely a handful of code lines per method. Our algorithm (abbreviated GTTF, Section 3.2), receives **g**raphs as input, **t**raverses them using efficient **t**ensor[1] operations, and invokes specializable **f**unctions during the traversal. We show function specializations for recovering popular GRL methods (Section 3.3). Moreover, since GTTF is stochastic, these specializations automatically scale to arbitrarily large graphs, without careful derivation per method. Importantly, such specializations, in expectation, recover unbiased gradient estimates of the objective w.r.t. model parameters.

---

[1]To disambiguate: by *tensors*, we refer to multi-dimensional arrays, as used in Deep Learning literature; and by *operations*, we refer to routines such as matrix multiplication, advanced indexing, etc

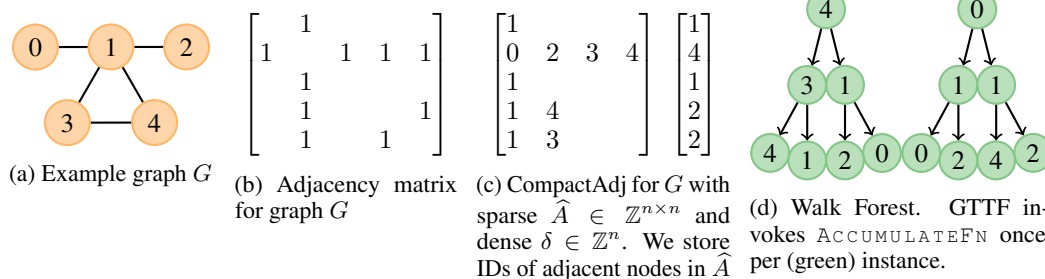

(a) Example graph $G$

(b) Adjacency matrix for graph $G$

(c) CompactAdj for $G$ with sparse $\widehat{A} \in \mathbb{Z}^{n \times n}$ and dense $\delta \in \mathbb{Z}^n$. We store IDs of adjacent nodes in $\widehat{A}$

(d) Walk Forest. GTTF invokes ACCUMULATEFN once per (green) instance.

Figure 1: (c)&(d) Depict our data structure & traversal algorithm on a toy graph in (a)&(b).

GTTF uses a data structure $\widehat{A}$ (*Compact Adjacency*, Section 3.1): a sparse encoding of the adjacency matrix. Node $v$ contains its neighbors in row $\widehat{A}[v] \triangleq \widehat{A}_v$, notably, in the first $degree(v)$ columns of $\widehat{A}[v]$. This encoding allows stochastic graph traversals using standard tensor operations. GTTF is a *functional*, as it accepts functions ACCUMULATEFN and BIASFN, respectively, to be provided by each GRL specialization to accumulate necessary information for computing the objective, and optionally to parametrize sampling procedure $p(v\text{'s neighbors} \mid v)$. The traversal internally constructs a *walk forest* as part of the computation graph. Figure 1 depicts the data structure and the computation. From a generalization perspective, GTTF shares similarities with Dropout (Srivastava et al., 2014).

Our contributions are: (i) A stochastic graph traversal algorithm (GTTF) based on tensor operations that inherits the benefits of vectorized computation and libraries such as PyTorch and Tensorflow. (ii) We list specialization functions, allowing GTTF to approximately recover the learning of a broad class of popular GRL methods. (iii) We prove that this learning is unbiased, with controllable variance. Wor this class of methods, (iv) we show that GTTF can scale previously-unscalable GRL algorithms, setting the state-of-the-art on a range of datasets. Finally, (v) we open-source GTTF along with new stochastic traversal versions of several algorithms, to aid practitioners from various fields in applying and designing state-of-the-art GRL methods for large graphs.

## 2 RELATED WORK

We take a broad standpoint in summarizing related work to motivate our contribution.

| Method | Family | Scale | Learning |
|---|---|---|---|
| *Models* | | | |
| GCN, GAT | MP | ✗ | exact |
| node2vec | NE | ✓ | approx |
| WYS | NE | ✗ | exact |
| *Stochastic Sampling Methods* | | | |
| SAGE | MP | ✓ | approx |
| FastGCN | MP | ✓ | approx |
| LADIES | MP | ✓ | approx |
| GraphSAINT | MP | ✓ | approx |
| CluterGCN | MP | ✓ | heuristic |
| *Software Frameworks* | | | |
| PyG | Both | | inherits / re- |
| DGL | Both | | implements |
| *Algorithmic Abstraction (ours)* | | | |
| GTTF | Both | ✓ | approx |

**Models** for GRL have been proposed, including *message passing* (**MP**) algorithms, such as Graph Convolutional Network (GCN) (Kipf & Welling, 2017), Graph Attention (GAT) (Veličković et al., 2018); as well as *node embedding* (**NE**) algorithms, including node2vec (Grover & Leskovec, 2016), WYS (Abu-El-Haija et al., 2018); among many others (Xu et al., 2018; Wu et al., 2019; Perozzi et al., 2014). The full-batch GCN of Kipf & Welling (2017), which drew recent attention and has motivated many MP algorithms, was not initially scalable to large graphs, as it processes all graph nodes at every training step. To scale MP methods to large graphs, researchers proposed **Stochastic Sampling Methods** that, at each training step, assemble a batch constituting subgraph(s) of the (large) input graph. Some of these sampling methods yield unbiased gradient estimates (with some variance) including SAGE (Hamilton et al., 2017), FastGCN (Chen et al., 2018b), LADIES (Zou et al., 2019), and GraphSAINT (Zeng et al., 2020). On the other hand, ClusterGCN (Chiang et al., 2019) is a heuristic in the sense that, despite its good performance, it provides no guarantee of unbiased gradient estimates of the full-batch learning. Gilmer et al. (2017) and Chami et al. (2021) generalized many GRL models into Message Passing and Auto-Encoder frameworks.

These frameworks prompt bundling of GRL methods under **Software Libraries**, like PyG (Fey & Lenssen, 2019) and DGL (Wang et al., 2019), offering consistent interfaces on data formats.

We now position our contribution relative to the above. Unlike generalized message passing (Gilmer et al., 2017), rather than abstracting the model computation, we abstract the learning algorithm. As a result, GTTF can be specialized to recover the learning of MP as well as NE methods. Morever, unlike Software Frameworks, which are re-implementations of many algorithms and therefore inherit the scale and learning of the copied algorithms, we re-write the algorithms themselves, giving them new properties (memory and computation complexity), while maintaining (in expectation) the original algorithm outcomes. Further, while the listed Stochastic Sampling Methods target MP algorithms (such as GCN, GAT, alike), as their initial construction could not scale to large graphs, our learning algorithm applies to a wider class of GRL methods, additionally encapsulating NE methods. Finally, while some NE methods such as node2vec (Grover & Leskovec, 2016) and DeepWalk (Perozzi et al., 2014) are scalable in their original form, their scalability stems from their multi-step process: sample many (short) random walks, save them to desk, and then learn node embeddings using positional embedding methods (e.g., word2vec, Mikolov et al. (2013)) – they are sub-optimal in the sense that their first step (walk sampling) takes considerable time (before training even starts) and also places an artificial limit on the number of training samples (number of simulated walks), whereas our algorithm conducts walks on-the-fly whilst training.

## 3 GRAPH TRAVERSAL VIA TENSOR FUNCTIONALS (GTTF)

At its core, GTTF is a stochastic algorithm that recursively conducts graph traversals to build representations of the graph. We describe the data structure and traversal algorithm below, using the following notation. $G = (V, E)$ is an unweighted graph with $n = |V|$ nodes and $m = |E|$ edges, described as a sparse adjacency matrix $A \in \{0,1\}^{n \times n}$. Without loss of generality, let the nodes be zero-based numbered i.e. $V = \{0, \ldots, n-1\}$. We denote the out-degree vector $\delta \in \mathbb{Z}^n$ – it can be calculated by summing over rows of $A$ as $\delta_u = \sum_{v \in V} A[u, v]$. We assume $\delta_u > 0$ for all $u \in V$: pre-processing can add self-connections to orphan nodes. $B$ denotes a batch of nodes.

### 3.1 DATA STRUCTURE

Internally, GTTF relies on a reformulation of the adjacency matrix, which we term *CompactAdj* (for "Compact Adjacency", Figure 1c). It consists of two tensors:

1. $\delta \in \mathbb{Z}^n$, a dense out-degree vector (figure 1c, right)
2. $\widehat{A} \in \mathbb{Z}^{n \times n}$, a sparse edge-list matrix in which the row $u$ contains left-aligned $\delta_u$ non-zero values. The consecutive entries $\{\widehat{A}[u, 0], \widehat{A}[u, 1], \ldots, \widehat{A}[u, \delta_u - 1]\}$ contain IDs of nodes receiving an edge from node $u$. The remaining $|V| - \delta_u$ are left unset, therefore, $\widehat{A}$ only occupies $\mathcal{O}(m)$ memory when stored as a sparse matrix (Figure 1c, left).

CompactAdj allows us to concisely describe stochastic traversals using standard tensor operations. To uniformly sample a neighbor to node $u \in V$, one can draw $r \sim \mathcal{U}[0..(\delta_u - 1)]$, then get the neighbor ID with $\widehat{A}[u, r]$. In vectorized form, given node batch $B$ and access to continuous $\mathcal{U}[0, 1)$, we sample neighbors for each node in $B$ as: $R \sim \mathcal{U}[0, 1)^b$, where $b = |B|$, then $B' = \widehat{A}[B, \lfloor R \circ \delta[B] \rfloor]$ is a $b$-sized vector, with $B'_u$ containing a neighbor of $B_u$, floor operation $\lfloor . \rfloor$ is applied element-wise, and $\circ$ is Hadamard product.

### 3.2 STOCHASTIC TRAVERSAL FUNCTIONAL ALGORITHM

Our traversal algorithm starts from a batch of nodes. It expands from each into a tree, resulting in a *walk forest* rooted at the nodes in the batch, as depicted in Figure 1d. In particular, given a node batch $B$, the algorithm instantiates $|B|$ *seed* walkers, placing one at every node in $B$. Iteratively, each walker first replicates itself a *fanout* ($f$) number of times. Each replica then samples and transitions to a neighbor. This process repeats a *depth* ($h$) number of times. Therefore, each seed walker becomes the ancestor of a $f$-ary tree with height $h$. Setting $f = 1$ recovers traditional random walk. In practice, we provide flexibility by allowing a custom fanout value per depth.

---

**Algorithm 1:** Stochastic Traverse Functional, parametrized by ACCUMULATEFN and BIASFN.

**input:** $u$ (current node); $T \leftarrow$ [] (path leading to $u$, starts empty); $F$ (list of fanouts);
ACCUMULATEFN (function: with side-effects and no return. It is model-specific and
records information for computing model and/or objective, see text);
BIASFN $\leftarrow \mathcal{U}$ (function mapping $u$ to distribution on $u$'s neighbors, defaults to uniform)

1   **def** *Traverse(T, u, F,* ACCUMULATEFN, BIASFN*)*:
2     **if** $F$.size() $= 0$ **then** return   # Base case. Traversed up-to requested depth
3     $f \leftarrow F$.pop()   # fanout duplication factor (i.e. breadth) at this depth.
4     sample_bias $\leftarrow$ BIASFN$(T, u)$
5     **if** sample_bias.sum() $= 0$ **then** return   # Special case. No sampling from zero mass
6     sample_bias $\leftarrow$ sample_bias / sample_bias.sum()   # valid distribution
7     $K \leftarrow$ *Sample*$(\widehat{A}[u, :\delta_u]]$, sample_bias$, f)$   # Sample $f$ nodes from $u$'s neighbors
8     **for** $k \leftarrow 0$ **to** $f - 1$ **do**
9       $T_{\text{next}} \leftarrow$ concatenate$(T, [u])$
10      ACCUMULATEFN$(T_{\text{next}}, K[k], f)$
11      *Traverse*$(T_{\text{next}}, K[k], f,$ ACCUMULATEFN, BIASFN$)$   # Recursion
12
13   **def** *Sample(N, W, f)*:
14     $C \leftarrow$ tf.cumsum$(W)$   # Cumulative sum. Last entry must = 1.
15     coin_flips $\leftarrow$ tf.random.uniform$((f, ), 0, 1)$
16     indices $\leftarrow$ tf.searchsorted$(C,$ coin_flips$)$
17     **return** $N[\text{indices}]$

---

Functional *Traverse* is listed in Algorithm 1. It accepts: a batch of nodes[2]; a list of fanout values $F$ (e.g. to $F = [3, 5]$ samples 3 neighbors per $u \in B$, then 5 neighbors for each of those); and more notably, two functions: ACCUMULATEFN and BIASFN. These functions will be called by the functional on every node visited along the traversal, and will be passed relevant information (e.g. the path taken from root seed node). Custom settings of these functions allow recovering wide classes of graph learning methods. At a high-level, our functional can be used in the following manner:

1. Construct model & initialize parameters (e.g. to random). Define ACCUMULATEFN and BIASFN.
2. Repeat (many rounds):
    i. Reset *accumulation information* (from previous round) and then sample batch $B \subset V$.
    ii. Invoke *Traverse* on $(B,$ ACCUMULATEFN, BIASFN$)$, which invokes the FN's, allowing the first to *accumulate information* sufficient for running the model and estimating an objective.
    iii. Use accumulated information to: run model, estimate objective, apply learning rule (e.g. SGD).

ACCUMULATEFN is a function that is used to track necessary information for computing the model and the objective function. For instance, an implementation of DeepWalk (Perozzi et al., 2014) on top of GTTF, specializes ACCUMULATEFN to measure an estimate of the sampled softmax likelihood of nodes' positional distribution, modeled as a dot-prodct of node embeddings. On the other hand, GCN (Kipf & Welling, 2017) on top of GTTF uses it to accumulate a sampled adjacency matrix, which it passes to the underlying model (e.g. 2-layer GCN) as if this were the *full* adjacency matrix.

BIASFN is a function that customizes the sampling procedure for the stochastic transitions. If provided, it must yield a probability distribution over nodes, given the current node and the path that lead to it. If not provided, it defaults to $\mathcal{U}$, transitioning to any neighbor with equal probability. It can be defined to read edge weights, if they denote importance, or more intricately, used to parameterize a second order Markov Chain (Grover & Leskovec, 2016), or use neighborhood attention to guide sampling (Veličković et al., 2018), as discussed in the Appendix.

---

[2]Our pseudo-code displays the traversal starting from **one node** rather than a batch only for clarity, as our actual implementation is vectorized e.g. $u$ would be a vector of nodes, $T$ would be a 2D matrix with each row containing transition path preceeding the corresponding entry in $u$, ... etc. Refer to Appendix and code.

### 3.3 SOME SPECIALIZATIONS OF ACCUMULATEFN & BIASFN

#### 3.3.1 MESSAGE PASSING: GRAPH CONVOLUTIONAL VARIANTS

These methods, including (Kipf & Welling, 2017; Hamilton et al., 2017; Wu et al., 2019; Abu-El-Haija et al., 2019; Xu et al., 2018) can be approximated by by initializing $\widetilde{A}$ to an empty sparse $n \times n$ matrix, then invoking *Traverse* (Algorithm 1) with $u = B$; $F$ to list of fanouts with size $h$; Thus ACCUMULATEFN and BIASFN become:

$$\textbf{def } \textsc{RootedAdjAcc}(T, u, f)\text{: } \widetilde{A}[u, T_{-1}] \leftarrow 1; \tag{1}$$

$$\textbf{def } \textsc{NoRevisitBias}(T, u)\text{: } \textbf{return } \mathbb{1}[\widetilde{A}[u].\text{sum}() = 0]\frac{\vec{\mathbf{1}}_{\delta_u}}{\delta_u}; \tag{2}$$

where $\vec{\mathbf{1}}_n$ is an $n$-dimensional all-ones vector, and negative indexing $T_{-k}$ is the $k^{\text{th}}$ last entry of $T$. If a node has been visited through the stochastic traversal, then it already has *fanout* number of neighbors and NoRevisitBias ensures it does not get revisited for efficiency, per line 5 of Algorithm 1. Afterwards, the accumulated stochastic $\widetilde{A}$ will be fed[3] into the underlying model e.g. for a 2-layer GCN of Kipf & Welling (2017):

$$\text{GCN}(\widetilde{A}, X; W_1, W_2) = \text{softmax}(\mathring{A} \text{ ReLu}(\mathring{A} X W_1) W_2); \qquad \overbrace{\phantom{xxxxxx}}^{\textit{renorm trick}} \tag{3}$$
$$\text{with } \mathring{A} = D'^{1/2}\widetilde{D}'^{-1}\widetilde{A}'D'^{-1/2}; \quad D' = \text{diag}(\delta'); \quad \delta' = \vec{\mathbf{1}}_n^\top A'; \quad \overbrace{\widetilde{A}' = I_{n \times n} + \widetilde{A}}$$

Lastly, $h$ should be set to the *receptive field* required by the model for obtaining output $d_L$-dimensional features at the labeled node batch. In particular, to the number of GC layers multiplied by the number of hops each layers access. E.g. hops=1 for GCN but customizable for MixHop and SimpleGCN.

#### 3.3.2 NODE EMBEDDINGS

Given a batch of nodes $B \subseteq V$, DeepWalk[4] can be implemented in GTTF by first initializing loss $\mathcal{L}$ to the contrastive term estimating the partition function of log-softmax:

$$\mathcal{L} \leftarrow \sum_{u \in B} \log \mathbb{E}_{v \sim P_n(V)}\left[\exp(\langle Z_u, Z_v \rangle)\right], \tag{4}$$

where $\langle ., . \rangle$ is dot-product notation, $Z \in \mathbb{R}^{n \times d}$ is the trainable embedding matrix with $Z_i \in \mathbb{R}^d$ is $d$-dimensional embedding for node $u \in V$. In our experiments, we estimate the expectation by taking 5 samples and we set the negative distribution $P_n(V = v) \propto \delta_v^{\frac{3}{4}}$, following Mikolov et al. (2013).

The functional is invoked with no BIASFN and ACCUMULATEFN =

$$\textbf{def } \textsc{DeepWalkAcc}(T, u, f)\text{: } \mathcal{L} \leftarrow \mathcal{L} - \left\langle Z_i, \sum_{k=1}^{C_T} \eta_{[T_{-k}]}\left(\frac{C-k+1}{C}\right)Z_{T_{-k}} \right\rangle; \ \eta_{[u]} \leftarrow \frac{\eta_{[T_{-1}]}}{f}; \tag{5}$$

where hyperparameter $C$ indicates maximum window size (inherited from word2vec, Mikolov et al., 2013), in the summation on $k$ does not access invalid entries of $T$ as $C_T \triangleq \min(C, T.\text{size})$, the scalar fraction $\left(\frac{C-k+1}{C}\right)$ is inherited from context sampling of word2vec (Section 3.1 in Levy et al., 2015), and rederived for graph context by Abu-El-Haija et al. (2018), and $\eta_{[u]}$ stores a scalar per node on the traversal Walk Forest, which defaults to 1 for non-initialized entries, and is used as a correction term. DeepWalk conducts random walks (visualized as a straight line) whereas our walk tree has a branching factor of $f$. Setting fanout $f = 1$ recovers DeepWalk's simulation, though we found $f > 1$ outperforms within fewer iterations e.g. $f = 5$, within 1 epoch, outperforms DeepWalk's published implementation. Learning can be performed using the accumulated $\mathcal{L}$ as: $Z \leftarrow Z - \epsilon \nabla_Z \mathcal{L}$;

## 4 THEORETICAL ANALYSIS

Due to space limitations, we include the full proofs of all propositions in the appendix.

---

[3]Before feeding the batch to model, in practice, we find nodes not reached by traversal and remove their corresponding rows (and also columns) from $X$ (and $A$).

[4]We present more methods in the Appendix.

## 4.1 ESTIMATING $k^{\text{TH}}$ POWER OF TRANSITION MATRIX

We show that it is possible with GTTF to accumulate an estimate of transition $\mathcal{T}$ matrix to power $k$. Let $\Omega$ denote the *walk forest* generated by GTTF, $\Omega(u, k, i)$ as the $i^{th}$ node in the vector of nodes at depth $k$ of the walk tree rooted at $u \in B$, and $t_i^{u,v,k}$ as the indicator random variable $\mathbb{1}[\Omega(u, k, i) = v]$. Let the estimate of the $k^{th}$ power of the transition matrix be denoted $\widehat{\mathcal{T}}^k$. Entry $\widehat{\mathcal{T}}_{u,v}^k$ should be an unbiased estimate of $\mathcal{T}_{u,v}^k$ for $u \in B$, with controllable variance. We define:

$$\widehat{\mathcal{T}}_{u,v}^k = \frac{\sum_{i=1}^{f^k} t_i^{u,v,k}}{f^k} \tag{6}$$

The fraction in Equation 6 counts the number of times the walker starting at $u$ visits $v$ in $\Omega$, divided by the total number of nodes visited at the $k^{\text{th}}$ step from $u$.

**Proposition 1.** (UNBIASEDTK) $\widehat{\mathcal{T}}_{u,v}^k$ *as defined in Equation 6, is an unbiased estimator of* $\mathcal{T}_{u,v}^k$

**Proposition 2.** (VARIANCETK) *Variance of our estimate is upper-bounded:* $Var[\widehat{\mathcal{T}}_{u,v}^k] \leq \dfrac{1}{4f^k}$

Naive computation of $k^{th}$ powers of the transition matrix can be efficiently computed via repeated sparse matrix-vector multiplication. Specifically, each column of $\mathcal{T}^k$ can be computed in $\mathcal{O}(mk)$, where $m$ is the number of edges in the graph. Thus, computing $\mathcal{T}^k$ in its entirety can be accomplished in $\mathcal{O}(nmk)$. However, this can still become prohibitively expensive if the graph grows beyond a certain size. GTTF on the other hand can estimate $\mathcal{T}^k$ in time complexity independent of the size of the graph, (Prop. 8), with low variance. Transition matrix powers are useful for many GRL methods. (Qiu et al., 2018)

## 4.2 UNBIASED LEARNING

As a consequence of Propositions 1 and 2, GTTF enables unbiased learning with variance control for classes of node embedding methods, and provides a convergence guarantee for graph convolution models under certain simplifying assumptions.

We start by analyzing node embedding methods. Specifically, we cover two general types. The first is based on matrix factorization of the power-series of transition matrix. and the second is based on cross-entropy objectives, e.g., like DeepWalk (Perozzi et al., 2014), node2vec (Grover & Leskovec, 2016), These two are shown in Proposations 3 and 4

**Proposition 3.** (UNBIASEDTFACTORIZATION) *Suppose* $\mathcal{L} = \frac{1}{2}||LR - \sum_k c_k \mathcal{T}^k||_F^2$, *i.e. factorization objective that can be optimized by gradient descent by calculating* $\nabla_{L,R}\mathcal{L}$, *where $c_k$'s are scalar coefficients. Let its estimate* $\widehat{\mathcal{L}} = \frac{1}{2}||LR - \sum_k c_k \widehat{\mathcal{T}}^k||_F^2$, *where $\widehat{\mathcal{T}}$ is obtained by GTTF according to Equation 6. Then* $\mathbb{E}[\nabla_{L,R}\widehat{\mathcal{L}}] = \nabla_{L,R}\mathcal{L}$.

**Proposition 4.** (UNBIASEDLEARNNE) *Learning node embeddings* $Z \in \mathbb{R}^{n \times d}$ *with objective function $\mathcal{L}$, decomposable as* $\mathcal{L}(Z) = \sum_{u \in V} \mathcal{L}_1(Z, u) - \sum_{u,v \in V} \sum_k \mathcal{L}_2(\mathcal{T}^k, u, v)\mathcal{L}_3(Z, u, v)$, *where $\mathcal{L}_2$ is linear over $\mathcal{T}^k$, then using $\widehat{\mathcal{T}}^k$ yields an unbiased estimate of* $\nabla_Z\mathcal{L}$.

Generally, $\mathcal{L}_1$ (and $\mathcal{L}_3$) score the similarity between disconnected (and connected) nodes $u$ and $v$. The above form of $\mathcal{L}$ covers a family of contrastive learning objectives that use cross-entropy loss and assume a logistic or (sampled-)softmax distributions. We provide, in the Appendix, the decompositions for the objectives of DeepWalk (Perozzi et al., 2014), node2vec (Grover & Leskovec, 2016) and WYS (Abu-El-Haija et al., 2018).

**Proposition 5.** (UNBIASEDMP) *Given input activations,* $H^{(l-1)}$, *graph conv layer* $(l)$ *can use rooted adjacency $\widetilde{A}$ accumulated by* ROOTEDADJACC *(1), to provide unbiased pre-activation output, i.e.* $\mathbb{E}\left[\mathring{A}^k H^{(l-1)} W^{(l)}\right] = \left(D'^{-1/2} A' D'^{-1/2}\right)^k H^{(l-1)} W^{(l)}$, *with $A'$ and $D'$ defined in (3).*

**Proposition 6.** (UNBIASEDLEARNMP) *If objective to a graph convolution model is convex and Lipschitz continous, with minimizer $\theta^*$, then utilizing GTTF for graph convolution converges to $\theta^*$.*

### 4.3 COMPLEXITY ANALYSIS

**Proposition 7.** STORAGE *complexity of GTTF is* $\mathcal{O}(m+n)$.

**Proposition 8.** TIME *complexity of GTTF is* $\mathcal{O}(bf^h)$ *for batch size $b$, fanout $f$, and depth $h$.*

Proposition 8 implies the speed of computation is irrespective of graph size. Methods implemented in GTTF inherit this advantage. For instance, the node embedding algorithm WYS (Abu-El-Haija et al., 2018) is $\mathcal{O}(n^3)$, however, we apply its GTTF implementation on large graphs.

## 5 EXPERIMENTS

We conduct experiments on 10 different graph datasets, listed in in Table 1. We experimentally demonstrate the following. (1) Re-implementing baseline method using GTTF maintains performance. (2) Previously-unscalable methods, can be made scalable when implemented in GTTF. (3) GTTF achieves good empirical performance when compared to other sampling-based approaches hand-designed for Message Passing. (4) GTTF consumes less memory and trains faster than other popular Software Frameworks for GRL. To replicate our experimental results, for each cell of the table in our code repository, we provide one shell script to produce the metric, except when we indicate that the metric is copied from another paper. Unless otherwise stated, we used fanout factor of 3 for GTTF implementations. Learning rates and model hyperparameters are included in the Appendix.

### 5.1 NODE EMBEDDINGS FOR LINK PREDICTION

In link prediction tasks, a graph is partially obstructed by hiding a portion of its edges. The task is to recover the hidden edges. We follow a popular approach to tackle this task: first learn node embedding $Z \in \mathbb{R}^{n \times d}$ from the observed graph, then predict the link between nodes $u$ and $v$ with score $\propto Z_u^\top Z_v$. We use two ranking metrics for evaluations: ROC-AUC, which is a ranking objective: how well do methods rank the hidden edges above randomly-sampled negative edges and Mean Rank.

We re-implement Node Embedding methods, DeepWalk (Perozzi et al., 2014) and WYS (Abu-El-Haija et al., 2018), into GTTF (abbreviated $\mathcal{F}$). Table 2 summarizes link prediction test performance.

LiveJournal and Reddit are large datasets, where original implementation of WYS is unable to scale to. However, scalable $\mathcal{F}$(WYS) sets new state-of-the-art on these datasets. For PPI and HepTh datasets, we copy accuracy numbers for DeepWalk and WYS from (Abu-El-Haija et al., 2018). For LiveJournal, we copy accuracy numbers for DeepWalk and PBG from (Lerer et al., 2019) – note that a well-engineered approach (PBG, (Lerer et al., 2019)), using a mapreduce-like framework, is under-performing compared to $\mathcal{F}$(WYS), which is a few lines specialization of GTTF.

### 5.2 MESSAGE PASSING FOR NODE CLASSIFICATION

We implement in GTTF the message passing models: GCN (Kipf & Welling, 2017), GraphSAGE (Hamilton et al., 2017), MixHop (Abu-El-Haija et al., 2019), SimpleGCN (Wu et al., 2019), as their computation is straight-forward. For GAT (Veličković et al., 2018) and GCNII (Chen et al., 2020), as they are more intricate, we download the authors' codes, and wrap them as-is with our functional.

We show that we are able to run these models in Table 3 (left and middle), and that GTTF implementations matches the baselines performance. For the left table, we copy numbers from the published papers. However, we update GAT to work with TensorFlow 2.0 and we use our updated code (GAT*).

### 5.3 EXPERIMENTS COMPARING AGAINST SAMPLING METHODS FOR MESSAGE PASSING

We now compare models trained with GTTF (where samples are walk forests) against sampling methods that are especially designed for Message Passing algorithms (GraphSAINT and ClusterGCN), especially since their sampling strategies do not match ours.

Table 3 (right) shows test performance on node classification accuracy on a large dataset: Products. We calculate the accuracy for $\mathcal{F}$(SAGE), but copy from (Hu et al., 2020) the accuracy for the baselines: GraphSAINT (Zeng et al., 2020) and ClusterGCN (Chiang et al., 2019) (both are message passing methods); and also node2vec (Grover & Leskovec, 2016) (node embedding method).

Table 1: Dataset summary. Tasks are LP, SSC, FSC, for link prediction, semi- and fully-supervised classification. Split indicates the train/validate/test paritioning, with (a) = (Abu-El-Haija et al., 2018), (b) = to be released, (c) = (Hamilton et al., 2017), (d) = (Yang et al., 2016); (e) = (Hu et al., 2020).

| Dataset | Split | # Nodes | # Edges | # Classes | Nodes | Edges | Tasks |
|---|---|---|---|---|---|---|---|
| PPI | (a) | 3,852 | 20,881 | N/A | proteins | interaction | LP |
| ca-HepTh | (a) | 80,638 | 24,827 | N/A | researchers | co-authorship | LP |
| ca-AstroPh | (a) | 17,903 | 197,031 | N/A | researchers | co-authorship | LP |
| LiveJournal | (b) | 4.85M | 68.99M | N/A | users | friendship | LP |
| Reddit | (c) | 233,965 | 11.60M | 41 | posts | user co-comment | LP/FSC |
| Amazon | (b) | 2.6M | 48.33M | 31 | products | co-purchased | FSC |
| Cora | (d) | 2,708 | 5,429 | 7 | articles | citation | SSC |
| CiteSeer | (d) | 3,327 | 4,732 | 6 | articles | citation | SSC |
| PubMed | (d) | 19,717 | 44,338 | 3 | articles | citation | SSC |
| Products | (e) | 2.45M | 61.86M | 47 | products | co-purchased | SSC |

Table 2: Results of node embeddings on Link Prediction. **Left**: Test ROC-AUC scores. **Right**: Mean Rank on the right for consistency with Lerer et al. (2019). *OOM = Out of Memory.

| | PPI | HepTh | Reddit |
|---|---|---|---|
| DeepWalk | 70.6 | 91.8 | 93.5 |
| $\mathcal{F}$(DeepWalk) | 87.9 | 89.9 | 95.5 |
| WYS | 89.8 | **93.6** | OOM |
| $\mathcal{F}$(WYS) | **90.5** | 93.5 | **98.6** |

| | LiveJournal |
|---|---|
| DeepWalk | 234.6 |
| PBG | 245.9 |
| WYS | OOM* |
| $\mathcal{F}$(WYS) | **185.6** |

Table 3: Node classification tasks. **Left**: test accuracy scores on semi-supervised classification (SSC) of citation networks. **Middle**: test micro-F1 scores for large fully-supervised classification. **Right**: test accuracy on an SSC task, showing only scalable baselines. We bold the highest value per column.

| | Cora | Citeseer | Pubmeb |
|---|---|---|---|
| GCN | 81.5 | 70.3 | 79.0 |
| $\mathcal{F}$(GCN) | 81.9 | 69.8 | 79.4 |
| MixHop | 81.9 | 71.4 | 80.8 |
| $\mathcal{F}$(MixHop) | 83.1 | 71.8 | **80.9** |
| GAT* | 83.2 | 72.4 | 77.7 |
| $\mathcal{F}$(GAT) | 83.3 | 72.5 | 77.8 |
| GCNII | **85.5** | 73.4 | 80.3 |
| $\mathcal{F}$(GCNII) | 85.3 | **74.4** | 80.2 |

| | Reddit | Amazon |
|---|---|---|
| SAGE | 95.0 | 88.3 |
| $\mathcal{F}$(SAGE) | **95.9** | 88.5 |
| SimpGCN | 94.9 | 83.4 |
| $\mathcal{F}$(SimpGCN) | 94.8 | **83.8** |

| | Products |
|---|---|
| node2vec | 72.1 |
| ClusterGCN | 75.2 |
| GraphSAINT | **77.3** |
| $\mathcal{F}$(SAGE) | 77.0 |

Table 4: Performance of GTTF against frameworks DGL and PyG. **Left**: Speed is the per epoch time in seconds when training GraphSAGE. Memory is the memory in GB used when training GCN. All experiments conducted using an AMD Ryzen 3 1200 Quad-Core CPU and an Nvidia GTX 1080Ti GPU. **Right**: Training curve for GTTF and PyG implementations of Node2Vec.

| | Speed (s) | | Memory (GB) | | | |
|---|---|---|---|---|---|---|
| | Reddit | Products | Reddit | Cora | Citeseer | Pubmed |
| DGL | 17.3 | 13.4 | OOM | 1.1 | 1.1 | 1.1 |
| PyG | 5.8 | 9.2 | OOM | 1.2 | 1.3 | 1.6 |
| GTTF | **1.8** | **1.4** | **2.44** | **0.32** | **0.40** | **0.43** |

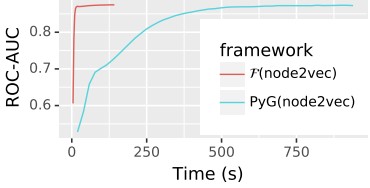

## 5.4 RUNTIME AND MEMORY COMPARISON AGAINST OPTIMIZED SOFTWARE FRAMEWORKS

In addition to the accuracy metrics discussed above, we also care about computational performance. We compare against software frameworks DGL (Wang et al., 2019) and PyG (Fey & Lenssen, 2019). These software frameworks offer implementations of many methods. Table 4 summarizes the following. First (left), we show time-per-epoch on large graphs of their implementation of

GraphSAGE, compared with GTTF's, where we make all hyper parameters to be the same (of model architecture, and number of neighbors at message passing layers). Second (middle), we run their GCN implementation on small datasets (Cora, Citeseer, Pubmed) to show peak memory usage. The run times between GTTF, PyG and DGL are similar for these datasets. The comparison can be found in the Appendix. While the aforementioned two comparisons are on popular message passing methods, the third (right) chart shows a popular node embedding method: node2vec's link prediction test ROC-AUC in relation to its training runtime.

## 6 CONCLUSION

We present a new algorithm, Graph Traversal via Tensor Functionals (GTTF) that can be specialized to re-implement the algorithms of various Graph Representation Learning methods. The specialization takes little effort per method, making it straight-forward to port existing methods or introduce new ones. Methods implemented in GTTF run efficiently as GTTF uses tensor operations to traverse graphs. In addition, the traversal is stochastic and therefore automatically makes the implementations scalable to large graphs. We theoretically show that the learning outcome due to the stochastic traversal is in expectation equivalent to the baseline when the graph is observed at-once, for popular GRL methods we analyze. Our thorough experimental evaluation confirms that methods implemented in GTTF maintain their empirical performance, and can be trained faster and using less memory even compared to software frameworks that have been thoroughly optimized.

## 7 ACKNOWLEDGEMENTS

We acknowledge support from the Defense Advanced Research Projects Agency (DARPA) under award FA8750-17-C-0106.

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

APPENDIX

## A    HYPERPARAMETERS

For the general link prediction tasks we used a $|B| = |V|$, $C = 5$, $f = 3$, 10 negative samples per edge, Adam optimizer with a learning rate of 0.5, multiplied by a factor of 0.2, every 50 steps, for 200 total iterations. The differences are listed below.

The Reddit dataset was trained using a starting learning rate of 2.0, decaying 50% every 10 iterations.

The LiveJournal task was trained using a fixed learning rate of 0.001, $|B| = 5000$, $f = 2$, and 50 negative samples per edge.

For the node classifications tasks:

For $\mathcal{F}(\text{SimpleGCN})$ on Amazon, we use $f = [15, 15]$, a batch size of 1024, and a learning rate of 0.02, decaying by a factor 0f 0.2 after 2 and 6 epochs for a total of 25 epochs. On Reddit, it is the same except $f = [25, 25]$

For $\mathcal{F}(\text{SAGE})$ on Amazon we use $f = [20, 10]$, a two layer model, a batch size of 256, and fixed learning rates of 0.001 and 0.002 respectively. On reddit we use $f = [25, 20]$, a fixed learning rate of 0.001, hidden dimension of 256 and a batch size of 1024. On the Products dataset, we used $f = [15, 10]$, a fixed learning learning rate of 0.001 and a batch size of 1024, a hidden dimension of 256 and a fixed learning rate of 0.003.

For GAT (baseline), we follow the authors code and hyperparameters: for Cora and Citeseer, we use Adam with learning rate of 0.005, L2 regularization of 0.0005, 8 attention heads on the first layer and 1 attention head on the output layer. For Pubmed, we use Adam with learning rate of 0.01, L2 regularization of 0.01, 8 attention heads on the first layer and 8 attention heads on the output layer. For $\mathcal{F}(\text{GAT})$, we use the same aforementioned hyperparameters, a fanout of 3 and traversal depth of 2 (to cover two layers) i.e. $F = [3, 3]$. For $\mathcal{F}(\text{GCN})$, we use the authors' recommended hyperparameters. Learning rate of 0.005, 0.001 L2 regularization, and $F = [3, 3]$, for all datasets. For both methods, we apply "patience" and stop the training if validation loss does not improve for 100 consecutive epochs, reporting the test accuracy at the best validation loss. For $\mathcal{F}(\text{MixHop})$, we wrap the authors' script and use their hyperparameters. For $\mathcal{F}(\text{GCNII})$, we use $F = [5, 5, 5, 5, 5, 5]$, as their models are deep (64 layers for Cora). Otherwise, we inherit their network hyperparameters (latent dimensions, number of layers, dropout factor, and their introduced coefficients), as they have tuned them per dataset, but we change the learning rate to $0.005$ (half of what they use) and we extend the patience from 100 to 1000, and extend the maximum number of epochs from 1500 to 5000 – this is because we are presenting a subgraph at each epoch, and therefore we intuitively want to slow down the learning per epoch, which is similar to the practice when someone applies Dropout to a neural networks. We re-run their shell scripts, with their code modified to use the Rooted Adjacency rather than the real adjacency, which is sampled at every epoch.

MLP was trained with 1 layer and a learning rate of 0.01.

# B PROOFS

## B.1 PROOF OF PROPOSITION 1

*Proof.* $\mathbb{E}[\widehat{\mathcal{T}}_{u,v}^k] = \mathbb{E}\left[\frac{\sum_{i=1}^{f^k} t_i^{u,v,k}}{f^k}\right] = \frac{\sum_{i=1}^{f^k} \mathbb{E}[t_i^{u,v,k}]}{f^k} = \frac{\sum_{i=1}^{f^k} P[t_i^{u,v,k} = 1]}{f^k} = \frac{\sum_{i=1}^{f^k} \mathcal{T}_{u,v}^k}{f^k} = \mathcal{T}_{u,v}^k$ □

## B.2 PROOF OF PROPOSITION 2

*Proof.* $\mathrm{Var}[\widehat{\mathcal{T}}_{u,v}^k] = \frac{\sum_{i=1}^{f^k} Var[t_i^{u,v,k}]}{f^{2k}} = \frac{f^k \mathcal{T}_{u,v}^k(1 - \mathcal{T}_{u,v}^k)}{f^{2k}} = \frac{\mathcal{T}_{u,v}^k(1 - \mathcal{T}_{u,v}^k)}{f^k}$

Since $0 \le \mathcal{T}_{u,v}^k \le 1$, then $\mathcal{T}_{u,v}^k(1 - \mathcal{T}_{u,v}^k)$ is maximized with $\mathcal{T}_{u,v}^k = \frac{1}{2}$. Hence $Var[\widehat{\mathcal{T}}_{u,v}^k] \le \frac{1}{4f^k}$ □

## B.3 PROOF OF PROPOSITION 3

*Proof.* Consider a $d$-dimensional factorization of $\sum_k c_k \mathcal{T}^k$, where $c_k$'s are scalar coefficients:

$$\mathcal{L} = \frac{1}{2}\left\|LR - \sum_k c_k \mathcal{T}^k\right\|_F^2, \tag{7}$$

parametrized by $L, R^\top \in \mathbb{R}^{n \times d}$. The gradients of $\mathcal{L}$ w.r.t. parameters are:

$$\nabla_L \mathcal{L} = \left(LR^\top - \sum_k c_k \mathcal{T}^k\right)R^\top \quad \text{and} \quad \nabla_R \mathcal{L} = L^\top\left(LR^\top - \sum_k c_k \mathcal{T}^k\right). \tag{8}$$

Given estimate objective $\mathcal{L}$ (replacing $\widehat{\mathcal{T}}$ with using GTTF-estimated $\widehat{\mathcal{T}}$):

$$\widehat{\mathcal{L}} = \frac{1}{2}\left\|LR - \sum_k c_k \widehat{\mathcal{T}}^k\right\|_F^2. \tag{9}$$

It follows that:

$$\begin{aligned}
\mathbb{E}\left[\nabla_L \widehat{\mathcal{L}}\right] &= \mathbb{E}\left[\left(LR^\top - \sum_k c_k \widehat{\mathcal{T}}^k\right)R^\top\right] \\
&= \mathbb{E}\left[\left(LR^\top - \sum_k c_k \widehat{\mathcal{T}}^k\right)\right]R^\top && \text{Scaling property of expectation} \\
&= \left(LR^\top - \sum_k c_k \mathbb{E}\left[\widehat{\mathcal{T}}^k\right]\right)R^\top && \text{Linearity of expectation} \\
&= \left(LR^\top - \sum_k c_k \mathcal{T}^k\right)R^\top && \text{Proof of Proposition 1} \\
&= \nabla_L \mathcal{L}
\end{aligned}$$

The above steps can similarly be used to show $\mathbb{E}\left[\nabla_R \widehat{\mathcal{L}}\right] = \nabla_R \mathcal{L}$ □

## B.4 PROOF OF PROPOSITION 4

*Proof.* We want to show that $\mathbb{E}[\nabla_Z \mathcal{L}(\widehat{\mathcal{T}}^k, Z)] = \nabla_Z \mathcal{L}(\mathcal{T}^k, Z)$. Since the terms of $\mathcal{L}_1$ are unaffected by $\widehat{\mathcal{T}}$, they are excluded w.l.g. from $\mathcal{L}$ in the proof.

$$\mathbb{E}[\nabla_Z \mathcal{L}(\widehat{\mathcal{T}}^k, Z)] = \mathbb{E}\left[\nabla_Z\left(-\sum_{u,v \in V}\sum_{k \in \{1..C\}} \mathcal{L}_2(\widehat{\mathcal{T}}^k, u, v)\mathcal{L}_3(Z, u, v)\right)\right]$$

$$\text{(by linearity of expectation)} = -\nabla_Z \sum_{u,v \in V} \sum_{k \in \{1..C\}} \mathcal{L}_2(\mathbb{E}[\widehat{\mathcal{T}}^k], u, v) \mathcal{L}_3(Z, u, v)$$

$$\text{(by Prop 1)} = -\nabla_Z \sum_{u,v \in V} \sum_{k \in \{1..C\}} \mathcal{L}_2(\mathcal{T}^k, u, v) \mathcal{L}_3(Z, u, v) = \nabla_Z \mathcal{L}(\mathcal{T}^k, Z)$$

$\square$

The following table gives the decomposition for DeepWalk, node2vec, and Watch Your Step. Node2vec also introduces a biased sampling procedure based on hyperparameters (they name $p$ and $q$) instead of uniform transition probabilities. We can equivalently bias the transitions in GTTF to match node2vec's. This would then show up as a change in $\widehat{\mathcal{T}}^k$ in the objective. This effect can also be included in the objective by multiplying $\langle Z_u, Z_v \rangle$ by the probability of such a transition in $\mathcal{L}_3$. In this format, the $p$ and $q$ variables appear in the objective and can be included in the optimization. For WYS, $Q_k$ are also trainable parameters.

| Method | $\mathcal{L}_1$ | $\mathcal{L}_2$ | $\mathcal{L}_3$ |
|---|---|---|---|
| DeepWalk | $0$ | $\left(\dfrac{C-k+1}{C}\right)\mathcal{T}^k$ | $\log(\langle Z_u, Z_v \rangle)$ |
| Node2Vec | $\log\left(\sum_{v \in V} \exp(\langle Z_u, Z_v \rangle)\right)$ | $\left(\dfrac{C-k+1}{C}\right)\mathcal{T}^k$ | $\langle Z_u, Z_v \rangle$ |
| Watch Your Step | $\log(1 - \sigma(\langle L_u, R_v \rangle))$ | $Q_k \mathcal{T}^k$ | $\log(\sigma(\langle L_u, R_v \rangle))$ |

Table 5: Decomposition of graph embedding methods to demonstrate unbiased learning. For WYS, $Z_u = \text{concatenate}(L_u, R_u)$.

For methods in which the transition distribution is not uniform, such as node2vec, there are two options for incorporating this distribution in the loss. The obvious choice is to sample from a biased transition matrix, $\mathcal{T}_{u,v} = \widetilde{W}_{u,v}$, where $\widetilde{W}$ is the transition weights. Alternatively, the transition bias can be used as a weight on the objective itself. This approach is still unbiased as

$$\mathbb{E}_{v \sim \widetilde{W}_u}[\mathcal{L}(v, u)] = \sum_{v \in V} P_{v \sim \widetilde{W}_u}[v]\mathcal{L}(v, u) = \sum_{v \in V} \widetilde{W}_{u,v}\mathcal{L}(v, u)$$

### B.5 Proof of Proposition 5

*Proof.* Let $\widetilde{A}$ be the neighborhood patch returned by GTTF, and let $\widetilde{\phantom{i}}$ indicate a measurement based on the sampled graph, $\widetilde{A}$, such as the degree vector, $\widetilde{\delta}$, or diagonal degree matrix, $\widetilde{D}$. For the remainder of this proof, let all notation for adjacency matrices, $A$ or $\widetilde{A}$, and diagonal degree matrices, $D$ or $\widetilde{D}$, and degree vector, $\delta$, refer to the corresponding measure on the graph with self loops e.g. $A \leftarrow A + I_{n \times n}$. We now show that the expectation of the layer output is unbiased.

$$\mathbb{E}\left[\mathring{A}^k H^{(l-1)} W^{(l)}\right] = \left[\mathbb{E}\left[\mathring{A}^k\right] H^{(l-1)} W^{(l)}\right] \text{ implies that } \mathbb{E}\left[\mathring{A}^k H^{(l-1)} W^{(l)}\right] \text{ is unbiased if } \mathbb{E}\left[\mathring{A}^k\right] = \left(D^{-1/2} A D^{-1/2}\right)^k.$$

$$\mathbb{E}\left[\mathring{A}^k\right] = \mathbb{E}\left[D^{1/2}\left(\widetilde{D}^{-1}\widetilde{A}\right)^k D^{-1/2}\right] = D^{1/2}\mathbb{E}\left[\left(\widetilde{D}^{-1}\widetilde{A}\right)^k\right] D^{-1/2}$$

Let $\mathcal{P}^{u,v,k}$ be the set of all walks $\{p = (u, v_1, ..., v_{k-1}, v) | v_i \in V\}$, and let $p \exists \widetilde{A}$ indicate that the path $p$ exists in the graph given by $\widetilde{A}$. Let $t^{u,v,k}$ be the transition probability from $u$ to $v$ in $k$ steps, and let $t^p$ be the probability of a random walker traversing the graph along path $p$.

$$\mathbb{E}\left[\left(\widetilde{D}^{-1}\widetilde{A}\right)^k_{u,v}\right] = \mathbb{E}\left[\widetilde{\mathcal{T}}^k_{u,v}\right] = Pr\left[\widetilde{t}^{u,v,k} = 1\right] = \sum_{p \in \mathcal{P}^{u,v,k}} Pr[p \exists \widetilde{A}] Pr[\widetilde{t}^p = 1 | p \exists \widetilde{A}]$$

$$= \sum_{p \in \mathcal{P}^{u,v,k}} \prod_{i=1}^{k} \mathbb{1}[A[p_i, p_{i+1}] = 1] \frac{f+1}{\delta[p_i]} \prod_{i=1}^{k} (f+1)^{-1} = \sum_{p \in \mathcal{P}^{u,v,k}} \prod_{i=1}^{k} \mathbb{1}[A[p_i, p_{i+1}] = 1] \delta[p_i]^{-1}$$

$$= \sum_{p \in \mathcal{P}^{u,v,k}} Pr[t^p = 1] = Pr[t^{u,v,k}] = (\mathcal{T})_{u,v}^k = (D^{-1}A)_{u,v}^k$$

Thus, $\mathbb{E}\left[\mathring{A}^k\right] = \left(D^{-1/2}AD^{-1/2}\right)^k$ and $\mathbb{E}\left[\mathring{A}^k H^{(l-1)} W^{(l)}\right] = \left(D^{-1/2}AD^{-1/2}\right)^k H^{(l-1)} W^{(l)}$

$\square$

For writing, we assumed nodes have degree, $\delta_u \geq f$, though the proof still holds if that is not the case as the probability of an outgoing edge being present from $u$ becomes 1 and the transition probability becomes $\delta_u^{-1}$ i.e. the same as no estimate at all.

### B.6 PROOF OF PROPOSITION 6

GTTF can be seen as a way of applying dropout (Srivastava et al., 2014), and our proof is contingent on the convergence of dropout, which is shown in Baldi & Sadowski (2014). Our dropout is on the adjacency, rather than the features. Denote the output of a graph convolution network[5] with $H$:

$$H = \text{GCN}_X(A; W) = \mathcal{T}XW$$

We restrict our analysis to GCNs with linear activations. We are interested in quantifying the change of $H$ as $A$ changes, and therefore the fixed (always visible) features $X$ is placed on the subscript. Let $\widetilde{A}$ denote adjacency accumulated by GTTF's ROOTEDADJACC (Eq. 1).

$$\widetilde{H}_c = \text{GCN}_X(\widetilde{A}_c).$$

Let $\mathcal{A} = \{\widetilde{A}_c\}_{c=1}^{|\mathcal{A}|}$ denote the (countable) set of all adjacency matrices realizable by GTTF. For the analysis, assume the graph is $\alpha$-regular: the assumption eases the notation though it is not needed. Therefore, degree $\delta_u = \alpha$ for all $u \in V$. Our analysis depends[6] on $\frac{1}{|\mathcal{A}|} \sum_{\widetilde{A} \in \mathcal{A}} \widetilde{A} \propto A$. i.e. the average realizable matrix by GTTF is proportional (entry-wise) to the full adjacency. This is can be shown when considering one-row at a time: given node $u$ with $\delta_u = \alpha$ outgoing neighbors, each of its neighbors has the same appearance probability $= \frac{1}{\delta_u}$. Summing over all combinations $\binom{\delta_u}{f}$, makes each edge appear the same frequency $= \frac{1}{\delta_u}|\mathcal{A}|$, noting that $|\mathcal{A}|$ evenly divides $\binom{\delta_u}{f}$ for all $u \in V$.

We define a dropout module:

$$\mathring{A} = \sum_{c}^{|\mathcal{A}|} z_c \widetilde{A}_c \quad \text{with} \quad z \sim \text{Categorical}\left(\overbrace{\frac{1}{|\mathcal{A}|}, \frac{1}{|\mathcal{A}|}, \cdots, \frac{1}{|\mathcal{A}|}}^{|\mathcal{A}| \text{ of them}}\right), \quad (10)$$

where $z_c$ acts as Multinoulli selector over the elements of $\mathcal{A}$, with one of its entries set to 1 and all others to zero. With this definitions, GCNs can be seen in the droupout framework as: $\widetilde{H} = \text{GCN}_X(\mathring{A})$. Nonetheless, in order to inherit the analysis of (Baldi & Sadowski, 2014, see their equations 140 & 141), we need to satisfy two conditions which their analysis is founded upon:

(i) $\mathbb{E}[\text{GCN}_X(\mathring{A})] = \text{GCN}_X(A)$: in the usual (feature-wise) dropout, such condition is easily verified.

(ii) Backpropagated error signal does not vary too much around around the mean, across all realizations of $\mathring{A}$.

Condition (i) is satisfied due to proof of Proposition 5. To analyze the error signal, i.e. the gradient of the error w.r.t. the network, assume loss function $\mathcal{L}(H)$, outputs scalar loss, is $\lambda$-Lipschitz continuous.

---

[5]The following definition averages the node features (uses non-symmetric normalization) and appears in multiple GCN's including Hamilton et al. (2017).

[6]If not $\alpha$-regular, it would be $\frac{1}{|\mathcal{A}|} \sum_{\widetilde{A} \in \mathcal{A}} \widetilde{A} \propto D^{-1}A$

The Liptchitz continuity allows us to bound the difference in error signal between $\mathcal{L}(H)$ and $\mathcal{L}(\widetilde{H})$:

$$||\boldsymbol{\nabla}_H\mathcal{L}(H) - \boldsymbol{\nabla}_H\mathcal{L}(\widetilde{H})||_2^2 \overset{\text{(a)}}{\leq} \lambda\left(\boldsymbol{\nabla}_H\mathcal{L}(H) - \boldsymbol{\nabla}_H\mathcal{L}(\widetilde{H})\right)^\top (H - \widetilde{H}) \tag{11}$$

$$\overset{\text{(b)}}{\leq} \lambda\,||\boldsymbol{\nabla}_H\mathcal{L}(H) - \boldsymbol{\nabla}_H\mathcal{L}(\widetilde{H})||_2\,||H - \widetilde{H}||_2 \tag{12}$$

$$\overset{\text{w.p. } \geq 1 - \frac{1}{Q^2}}{\leq} \lambda\,||\boldsymbol{\nabla}_H\mathcal{L}(H) - \boldsymbol{\nabla}_H\mathcal{L}(\widetilde{H})||_2\,W^\top X^\top Q\sqrt{Var[\mathcal{T}]}XW \tag{13}$$

$$= \frac{\lambda Q}{2\sqrt{f}}\,||\boldsymbol{\nabla}_H\mathcal{L}(H) - \boldsymbol{\nabla}_H\mathcal{L}(\widetilde{H})||_2\,||W||_1^2\,||X||_1^2 \tag{14}$$

$$||\boldsymbol{\nabla}_H\mathcal{L}(H) - \boldsymbol{\nabla}_H\mathcal{L}(\widetilde{H})||_2 \leq \frac{\lambda Q}{2\sqrt{f}}\,||W||_1^2\,||X||_1^2 \tag{15}$$

where (a) is by Lipschitz continuity, (b) is by Cauchy–Schwarz inequality, "w.p." means with probability and uses Chebyshev's inequality, with the following equality because the variance of $\mathcal{T}$ is shown element-wise in proof for Prop. 2. Finally, we get the last line by dividing both sides over the common term. This shows that one can make the error signal for the different realizations arbitrarily small, for example, by choosing a larger fanout value or putting (convex) norm constraints on $W$ and $X$ e.g. through batchnorm and/or weightnorm. Since we can have $\boldsymbol{\nabla}_H\mathcal{L}(H) \approx \boldsymbol{\nabla}_H\mathcal{L}(\widetilde{H}_1) \approx \boldsymbol{\nabla}_H\mathcal{L}(\widetilde{H}_2) \approx \cdots \approx \boldsymbol{\nabla}_H\mathcal{L}(\widetilde{H}_{|\mathcal{A}|})$ with high probability, then the analysis of Baldi & Sadowski (2014) applies. Effectively, it can be thought of as an online learning algorithm where the elements of $\mathcal{A}$ are the stochastic training examples and analyzed per (Bottou, 1998; 2004), as explained by Baldi & Sadowski (2014) $\square$.

### B.7 Proof of Proposition 7

The storage complexity of *CompactAdj* is $\mathcal{O}(sizeof(\delta) + sizeof(\widehat{A})) = \mathcal{O}(n + m)$.

Moreover, for extemely large graphs, the adjacncy can be row-wise partitioned across multiple machines and therefore admitting linear scaling. However, we acknolwedge that choosing which rows to partition to which machines can drastically affect the performance. Balanced partitioning is ideal. It is an NP-hard problem, but many approximations have been proposed. Nonetheless, reducing inter-communication, when distributing the data structure across machines, is outside our scope.

### B.8 Proof of Proposition 8

For each step of GTTF, the computational complexity is $\mathcal{O}(bh^f)$. This follows trivially from the GTTF functional: each nodes in batch ($b$ of them) builds a tree with depth $h$ and fanout $f$ i.e. with $h^f$ tree nodes. This calculation assumes random number generation, Accumulate­Fn and Bias­Fn take constant time. The `searchsorted` function is linear, as it is called on a sorted list: cumulative sum of probabilities.

## C Additional GTTF Implementations

### C.1 Message Passing Implementations

#### C.1.1 Graph Attention Networks (GAT, Veličković et al., 2018)

One can implement GAT by following the previous subsection, utilizing Accumulate­Fn and Bias­Fn defined in (1) and (2), but just replacing the model (3) by GAT's:

$$\text{GAT}(\widetilde{A}, X; \mathcal{A}, W_1, W_2) = \text{softmax}((\mathcal{A} \circ \mathring{A})\,\text{ReLu}((\mathcal{A} \circ \mathring{A})XW_1)W_2); \tag{16}$$

where $\circ$ is hadamard product and $\mathcal{A}$ is an $n \times n$ matrix placing a positive scalar (an attention value) on each edge, parametrized by multi-headed attention described in (Veličković et al., 2018). However, for some high-degree nodes that put most of the attention weight on a small subset of their neighbors, sampling uniformly (with Bias­Fn=NoRevisitBias) might mostly sample neighbors with entries in $\mathcal{A}$ with value $\approx 0$, and could require more epochs for convergence. However, our flexible functional

allows us to propose a sample-efficient alternative, that is in expectation, equivalent to the above:

$$\text{GAT}(\widetilde{A}, X; \mathcal{A}, W_1, W_2) = \text{softmax}((\sqrt{\mathcal{A}} \circ \mathring{A}) \text{ ReLu}((\sqrt{\mathcal{A}} \circ \mathring{A})XW_1)W_2); \qquad (17)$$

$$\textbf{def } \text{GATBIAS}(T, u): \textbf{return } \text{NOREVISITBIAS}(T, u) \circ \sqrt{\mathcal{A}[u, \widehat{A}[u]]}; \qquad (18)$$

### C.1.2 DEEP GRAPH INFOMAX (DGI, VELIČKOVIĆ ET AL., 2019)

DGI implementation on GTTF can use ACCUMULATEFN=ROOTEDADJACC, defined in (1). To create the positive graph: it can sample some nodes $B \subset V$. It would pass to GTTF's *Traverse B*, and utilize the accumulated adjacency $\widehat{A}$ for running: $\text{GCN}(\widehat{A}, X_B)$ and $\text{GCN}(\widehat{A}, X_{\text{permute}})$, where the second run randomly permutes the order of nodes in $X$. Finally, the output of those GCNs can then be fed into a readout function which outputs to a descriminator trying to classify if the readout latent vector correspond to the real, or the permuted features.

### C.2 NODE EMBEDDING IMPLEMENTATIONS

### C.2.1 NODE2VEC (GROVER & LESKOVEC, 2016)

A simple implementation follows from above: N2VACC $\triangleq$ DEEPWALKACC; but override BIASFN =

$$\textbf{def } \text{N2VBIAS}(T, u): \textbf{return } p^{-\mathbb{1}[i=T_{-2}]} q^{-\mathbb{1}[\langle A[T_{-2}], A[u]\rangle > 0]}; \qquad (19)$$

where $\mathbb{1}$ denotes indicator function, $p, q > 0$ are hyperparameters of node2vec assigning (unnormalized) probabilities for transitioning back to the previous node or to node connected to it. $\langle A[T_{-2}], A[u]\rangle$ counts mutual neighbors between considered node $u$ and previous $T_{-2}$.

An alternative implementation is to **not override** BIASFN but rather fold it into ACCUMULATEFN, as:

$$\textbf{def } \text{N2VACC}(T, u, f): \text{DEEPWALKACC}(T, u, f); \eta_{[u]} \leftarrow \eta_{[u]} \times \text{N2VBIAS}(T, u); \qquad (20)$$

Both alternatives are equivalent in expectation. However, the latter directly exposes the parameters $p$ and $q$ to the objective $\mathcal{L}$: allowing them to be differentiable w.r.t. $\mathcal{L}$ and therefore trainable via gradient descent, rather than by grid-search. Nonetheless, parameterizing $p$ & $q$ is beyond our scope.

### C.2.2 WATCH YOUR STEP (WYS, ABU-EL-HAIJA ET AL., 2018)

First, embedding dictionaries $R, L \in \mathbb{R}^{n \times \frac{d}{2}}$ can be initialized to random. Then repeatedly over batches $B \subseteq V$, the loss $\mathcal{L}$ can be initialized to estimate the negative part of the objective:

$$\mathcal{L} \leftarrow - \sum_{u \in B} \log \sigma(-\mathbb{E}_{v \in \mathcal{U}(V)} [\langle R_u, L_v \rangle + \langle R_v, L_u \rangle]),$$

Then call GTTF's traverse passing the following ACCUMULATEFN=

$$
\begin{aligned}
&\textbf{def } \text{WYSACC}(T, u): \\
&\quad \textbf{if } T.size() \neq Q.size(): \textbf{return}; \\
&\quad t \leftarrow T[0]; \quad U \leftarrow T[1:] \cup [u]; \\
&\quad \texttt{ctx\_weighted\_L} \leftarrow \sum_j Q_j L_{U_j}; \quad \texttt{ctx\_weighted\_R} \leftarrow \sum_j Q_j R_{U_j}; \\
&\quad \mathcal{L} \leftarrow \mathcal{L} - \log(\sigma(\langle R_t, \texttt{ctx\_weighted\_L}\rangle + \langle L_t, \texttt{ctx\_weighted\_R}\rangle));
\end{aligned}
$$

## D MISCELLANEOUS

### D.1 SENSITIVITY

The following figures show the sensitivity of fanout and walk depth for WYS on the Reddit dataset.

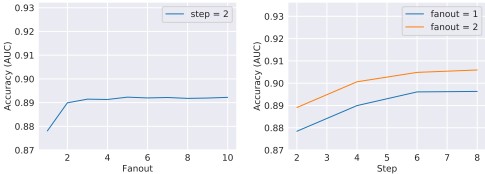

Figure 3: Test AUC score when changing the fanout (left) and random walk length (right)

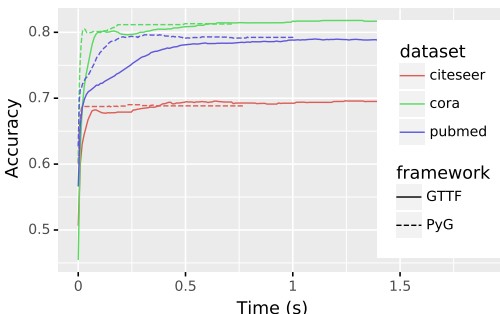

Figure 4: Runtime of GTTF versus PyG for training GCN on citation network datasets. Each line averages 10 runs on an Nvidia GTX 1080Ti GPU.

### D.2 RUNTIME OF GCN ON CITATION NETWORKS

Citation networks (Cora, Citeseer, Pubmed) have been popularized by Kipf & Welling (2017), and for completeness, we report in Figure 4 the runtime versus test accuracy of GCN on these networks. We compare against PyG, which is optimized for GCN. We find that the methods are somewhat comparable in terms of training time.

