# OpenReview forum: "Graph Traversal with Tensor Functionals: A Meta-Algorithm for Scalable Learning"
_ICLR.cc/2021/Conference — ICLR 2021 Poster_

### Official Review · AnonReviewer1 · 2020-10-21
**Authors proposed graph traversal via tensor functionals (GTTE) for easing the implementation of diverse graph algorithms and enabling transparent and efficient scaling to large graphs. It can be specialized to obtain various existing graph representation learning models.**

**Rating:** 7
**Confidence:** 3

**Review:**

Pros:
1)	The graph traversal via tensor functionals is proposed with stochastic graph traversal algorithm based on tensor optimization that inherits benefits of vectorized computation and libraries.
2)	It can be specialized for many existing models and make them scalable for large graph.
3)	Authors proved the learning is unbiased, with controllable variance.
4)	Experimental results on both node classification and link prediction verify the efficiency and effectiveness of the proposed unified meta-algorithm framework.

Cons:
1)	As shown in the experiments, the implementation of existing methods under GTTE can be more efficient and less memory requirement. It might be better to conduct complexity analysis on the proposed meta-algorithm to show these advantages.
2)	The algorithmic comparisons between the existing stochastic sampling methods and GTTE can be helpful to understand the differences and their pros and cons.

---

> ### Author Response · Authors · 2020-11-11
> **Complexity Analysis**
>
> Thank you for your time and clear review!
>
> In response to your feedback:
> * We do complexity analysis in section 4.3. We could create a table showing the complexity of other algorithms, if you’d like, to compare to GTTF (such as, other sampling methods or models implemented not under GTTF). In practice, GTTF is more useful for large datasets (as its runtime memory does not depend on size of graph) but we also wanted to highlight that it does not suffer accuracy even when applied to small datasets.
>  * While the sampling in GTTF bears similarity to other sampling methods, there are significant differences. As mentioned in the related work section, GTTF additionally generalizes to node embedding methods whereas sampling methods we are aware of focus only on message passing methods. Also, not all sampling methods are unbiased (e.g. ClusterGCN). Second, GTTF’s stochastic walk is implemented using fast vectorized operations, which were possible to implement given our Compact Adjacency. By customizing bias and accumulation functions, GTTF can recover both other sampling methods as well as other models. This is exemplified in how we recover the sampling from node2vec by modifying the bias function (Appendix C.1.2).

---

### Official Review · AnonReviewer4 · 2020-10-25
**A nice paper with wide range, suffering from a few problems**

**Rating:** 7
**Confidence:** 3

**Review:**

# Quality

This paper, as far as I can tell, is technically solid.
The authors present an algorithm, show how it can be specialized to a variety of other methods, then prove the conditions under which it estimates loss functions, gradients, and powers of transition matrices without bias.
They then demonstrate how their method can be used to approximate graph representation learning techniques over a variety of datasets, as well as enabling the application of non-scalable techniques to large graphs.

# Clarity

This paper is mostly well-written, with a few issues throughout.

One issue I took was with the authors' use of the term "tensor:" they claim that their method allows for efficient implementation via standard tensor operations.
This was deceiving when first reading the paper: I expected this approach to involve tensors in the sense of a multilinear operator, and for operations to be understood in terms of that operator.
The approach in this paper doesn't resemble tensor methods in that way: it would be better described as graph representation learning with sparse edge lists.

There is also some ambiguity in Algorithm 1: it is unclear what AccumulateFn *does*.
Without thoroughly looking at the specific applications, it is not easy to tell what effect line 10 in Algorithm 1 has.
It would significantly improve readability to be specific about side-effects, return values, etc. of AccumulateFn in general.

# Originality

This paper is somewhat original.
The authors do not formulate anything particularly new: rather, they unify a variety of algorithms for graph representation learning under a common approach, parameterized by two functions (Accumulate and Bias).
The algorithm in this paper takes very similar form to that written in the most popular GRL papers, so I would not evaluate it as being a particularly novel contribution.

However, there is novel value in the analysis of the meta-algorithm as an unbiased estimator of loss functions dependent on powers of a transition matrix.
This is similar in spirit to the cited paper [Qui et al., 2018], where a link is made between these methods and matrix factorization techniques.
Here, the authors have made a link between many methods and a general framework, from which they can form computationally efficient approximations.

# Significance

One contribution of this work to the community is a "unified look" at a wide array of node embedding methods.
The authors have identified common features of approaches such as node2vec, DeepWalk, GCN, etc., and formulated a stochastic approach that is flexible enough to reflect each of these algorithms.
I wouldn't characterize this formulation as inherently significant, since the form of all of these algorithms is written in a similar way.

A more concrete contribution of this work is the analysis of this formulation as a computationally efficient (Propositions 6 and 7) approximator for powers of a transition matrix (Propositions 1 and 2), which has applications in many representation learning tasks.
It is established that using the proposed estimator of the transition matrix yields unbiased estimates and gradients for a class of loss functions (Propositions 3, 4, and 5).

This yields the most significant contribution of this paper: if a practitioner can write the loss function in the form provided in Proposition 3, the formulation of the authors can be justified as a good, computationally efficient approximant for their case.
This impact is demonstrated in the experiments using WYS, where the authors' form is shown to perform similarly to WYS on graphs of small scale, and then shown to be feasible on graphs of large scale, while the original form of WYS does not scale.
That is, the proposed framework appears to be a reasonable way to approximate methods that are not scalable, showing potential in bridging theory and practice.

Beyond the special case where a method's loss function can be written in terms of a transition matrix raised to some power, though, this paper falls short.
It would be useful to understand the behavior of this approach when used to approximate methods that cannot be written in the form of Proposition 3.
Because of this, the scope of analysis in this paper is somewhat limited: the authors should more directly discuss cases where this is not true, beyond the brief mention of node2vec in the appendix.

# Overall review

This paper takes a broad view of graph representation learning problems, proposing a generic algorithm that they show can recover many popular approaches.
Although there are a few missing pieces, the proposed approach is useful in scaling approaches that were previously not scalable, and thus warrants being accepted into ICLR 2021.

---

> ### Author Response · Authors · 2020-11-11
> **Clarifications and further analysis on family of objective functions that GTTF can approximate**
>
> Thank you for the detailed feedback, especially in providing it in sections! We are echoing your sections in our response.
>
> ## Clarity
> We understand the confusion -- wikipedia says there are multiple common usages of “tensor”, including vector field (physics), muscles (anatomy/medicine), and our intended usage: multi-dimensional array (deep learning). Therefore, to clarify to readers, we added a footnote in the introduction: “To disambiguate: by *tensors*, we refer to multi-dimensional arrays, as used in Deep Learning literature; by *operations*, we refer to routines such as matrix multiplication, advanced indexing, etc”.
>
> The accumulate function will almost certainly have side-effects, but has no return value. The text has a description of what the accumulate function does (second to last paragraph, of Section 3.2) and how it is specialized (Section 3.3). Nonetheless, we modified the pseudo-code comment for the AccumulateFn in the algorithm section to help readers who jump into the algorithm section prior to reading the text: “function: with side-effects and no return. It is model-specific and records information for computing model and/or objective, see text” -- you may review in the revised paper.
>
>
> ## Significance
> The form we analyze is common in the literature. We added a sentence after Prop3:
> “The above form of $\mathcal{L}$ covers a family of contrastive learning objectives that use cross-entropy loss and assume a logistic or (sampled-)softmax distributions.”
>
> It is hard to do an analysis on the “universe of all objective functions involving A or T” Nonetheless, we just did back-of-the-envelope math on matrix factorization (decomposition) of $\mathcal{T}^k$. It is also unbiased! Specifically, let $L = \frac{1}{2} ||Q R - \mathcal{T}^k||^2_F$ be the true objective with parameters $Q, R^\top \in \mathbb{R}^{n \times d}$ and let $\widehat{L} = \frac{1}{2} ||Q R - \widehat{\mathcal{T}}^k||^2_F$ be an estimated objective where $\widehat{\mathcal{T}}^k$ is sampled using GTTF. We need to show that $\mathbb{E}[\partial  \widehat{L} / \partial Q] = \partial L / \partial Q$ and similarly for $R$. We know that $\frac{\partial L}{\partial Q} = (QR - \mathcal{T}^k) R^\top$ and that $\frac{\partial L}{\partial R} = Q^\top (QR - \mathcal{T}^k)$. It turns out, when analyzing $\mathbb{E}[d  \widehat{L} / d Q]$, we can push the expectation all the way onto $\widehat{\mathcal{T}}$:
> $\mathbb{E}[\partial \widehat{L} / \partial Q] = \mathbb{E}[(QR - \mathcal{T}^k) R^\top] = (QR - \mathbb{E}[\mathcal{T}^k]) R^\top = (QR - \mathcal{T}^k) R^\top $
> Where the last equality follows from Prop1 and the second-to-last follows from using linearity and scaling properties of expectation: https://www.randomservices.org/random/expect/Matrices.html. This can be extended to a linear combination of T^k (e.g. $\sum_k c_k \mathcal{T}^k$ and QR decomposition on adjacency A. We can add this analysis to the paper, if it gets accepted, as we would get a 9th page allowance.

---

### Official Review · AnonReviewer3 · 2020-10-28
**Good contribution but experiments are not convincing**

**Rating:** 7
**Confidence:** 2

**Review:**

This paper proposes a new graph traversal method by introducing a compact adjacency matrix and applying stochastic methods to sample nodes.  The proposed graph traversal can be applied in conjunction with graph neural networks such as GCN, Deepwalk,..etc, and ideas proposed in the paper would interest and influence other researchers in the community. An advantage is that the proposed method shows improvements in speed and memory usage. Overall, The paper is well written with both theoretical and experimental evaluations.

Pros:
a.) The proposed method well-developed and supported with theoretical analysis

b.) The proposed sampling method can be combined with other existing methods. The authors discuss the applicability of the proposed method with existing methods.

Cons:
a.) One limitation is that experiments are not very convincing since there are only a few baseline methods that are compared with the proposed method.  For instance, It would be better to show performances of methods like GraphSAGE, cluster GCN, and other methods (at least method in the table in Section 2) applied to all datasets. Showing different selections of baseline methods make it hard to understand the true performances of the proposed method. Can the authors provide more detailed comparisons with different baseline methods?

b.) The performances (e.g. semisupervised node-classification) shown are not as competitive as performances achieved by other recent methods (e.g. GCNII). Can the authors add more experimental results to the paper?

I raise my rating based on the additional experimental results given.

---

> ### Author Response · Authors · 2020-11-11
> **We ran GCNII under GTTF and we can add more experiments if appropriate**
>
> Thank you for recognizing the significance of the problem we are studying and also for your time and your insightful feedback.
>
> ## Experiments
>  * As our intention is to introduce a unifying algorithmic approach, rather than a new model, we do not expect to get “strong bold numbers” except in cases where we make a strong (but previously non-scalable model) to become scalable. Our intent is to show that re-implementation of the underlying methods do not show any performance losses. We acknowledge that having a table containing the cross product of all datasets against all models might look more convincing. However,  in our experiments we have tried to include the (model, dataset) combinations that are most relevant. For example, our initial experiments showed that SAGE performs worse on articles datasets (Cora, Citeseer, Pubmed), perhaps because these datasets are transductive whereas SAGE model design is geared towards inductive settings. Also it is not possible to run methods like GCN on large datasets such as Amazon and Reddit. Based on your suggestion, we would be willing to add a complete table to the appendix (and the table will have a number of OOM entries for some baselines not using GTTF).
>  * We were not aware of GCNII by the time we conducted the experiments (it looks like the paper made it to arxiv in July, around the time we were finalizing our experiments, we apologize for the omission). Nonetheless, we were able to wrap the author’s code with GTTF: Namely, we replaced their line 63 in their train.py (direct link: https://github.com/chennnM/GCNII/blob/master/train.py#L63) with one that feeds an adjacency sampled from GTTF, and we re-ran their code. We used faounts=[5, 5, 5, 5, 5, 5] i.e. only 6 hops. We were able to get their results (we see slight improvement on Cora, but almost the same results for datasets Pubmed and Citeseer - we currently average 5 runs, we will do more runs for consistency and re-update the paper). This gives us a hypothesis: while their model deep model is good for feature transformation (64 layers for cora, per their semi.sh), but does not really pool from 64 hops away (the diameter of these graphs is much smaller, anyway). We now have added the results into the experiments section (you may view updated result tables), as well as information on hyperparameters in the appendix

---

> > ### Comment · AnonReviewer3 · 2020-11-21
> > **Comment**
> >
> > Thank you for the reply! I think adding a detailed comparable table with different common baseline methods would be helpful for practitioners. Additionally to make the results more up to date I suggest that GCNII is also compared in the future versions of the paper.

---

> > > ### Author Response · Authors · 2020-11-22
> > > **RE: GCNII**
> > >
> > > GCNII experiment results are in the updated revision (you may view it, above). We will do runs this week to try all methods on all datasets and once they conclude, we will start a table in the appendix.

---

> > > ### Author Response · Authors · 2020-11-25
> > > **Detailed table in progress!**
> > >
> > > We've updated the code (software engineering-wise; maintaining the algorithm) and are now running all models on all datasets. We have prepared the shell scripts. Some have finished, while others will be queued.
> > >
> > > We think it would be great to include a full table of results, not only "in the usual setups" (E.g. SAGE is usually only run in inductive; GAT only on citations; etc) -- thank you for the idea. I think other readers might find it useful, too.
> > >
> > > Draft of this table has been added as a top-level comment, above.

---

### Official Review · AnonReviewer2 · 2020-10-28
**Approximating graph representation learning schemes via sampling random trees**

**Rating:** 7
**Confidence:** 3

**Review:**

### Summary

The authors propose a "meta-algorithm" for approximating various graph representation learning schemes: generate batches of random trees with fixed fanout (and possibly biased probabilities of selecting different edges), and use them to accumulate information to approximate operations on the graph.  The idea is beautifully simple, and generalizes running independent random walkers, an approach that is used in deriving many related algorithms.  The biasing and accumulation functions are user provided, and the authors show how different choices of these functions can be used to approximate a number of graph representation learning schemes.  The authors also provide a software framework, though it was inaccessible at review due to anonymization.  Experiments show the approach is much more scalable than competing approaches (though, to be fair, some of the competition was not targeting scalability).

### Minor points

- The "CompactAdj" format seems equivalent to compressed sparse row (or compressed sparse column) storage for the adjacency.  Is there any difference?
- It might be worth commenting (possibly in appendices) on the data structures used for accumulation in the different concrete instances that are presented.  For example, the sparse matrix data structure used in 3.3.1 is presumably a neighbor list or equivalent, and not the CompactAdj format (since rebuilding the latter at every step seems needlessly expensive).
- The claim at the end of section 4.1 that "although naive calculation of $T^k$ is cubic, GTTF can estimate it in time complexity independent of the number of nodes" is imprecise to the point of being misleading.  One can exactly compute a single row of $T^k$ for arbitrary $T$ via repeated sparse matrix-vector products time time proportional to the graph size times $k$, and one can do an approximate  computation of a row of $T^k$ even more cheaply using the method sketched here.  One can also get efficient estimates of various properties of matrix powers.  But actually getting estimates of all elements of the matrix $T^k$ in general is going to depend on the size of the graph, particularly when $k$ gets to the order of the graph diameter.

### Typos

- In the related work section, the table lists "DGI" -- I believe this is referring to DGL.
- For the paper, I recommend writing out "without loss of generality" rather than using the abbreviation at the start of section 3 (unless it leads to going over space)
- Add "the" before "node embedding WYS" at the start of 4.3.

### Update

Thanks to the authors for their clarifications (and to the other reviewers).  I am more comfortable with my accept recommendation now, and have updated my confidence accordingly.

---

> ### Author Response · Authors · 2020-11-11
> **Clarifying CompactAdj and correcting complexity for naive decomposition of T^K**
>
> Thank you for the nice words that our method is simple yet general. Our energy feeds on positive (and negative constructive) feedback, regardless of the final decision. Thank you, too, for the elaborate summary.
>
> ## Minor Points
>
>  * While similar to CSR format, Compact Adjacency is not the same. Namely, adjacency A is binary ({0, 1} entries) while CompactAdj stores node IDs ({0, …, n} entries) . Nonetheless, your intuition is correct: it is much faster to store our Compact Adjacency matrix in CSR format as, when visiting a node, we want access to its neighbors which live in the same row.
>
>  * Correct: the CompactAdj is built only once and stays fixed throughout training. Building it is linear in the edges (it takes  <1 second for small datasets, but up to a few minutes, for larger datasets). Our code has the option to cache it on disk (to save time when doing hyperparameter sweeps). For accumulation, the data structure is dependent on the algorithm being recovered and we will comment about the complexity of maintaining the data structure in the main paper (when we have more space). In the section you mention, the data structure is the true adjacency, but only marking edges visited during the walk forrest. Your guess is correct, our code maintains an adjacency list and converts at-once to a sparse matrix (to feed back into TensorFlow/PyTorch).
>
>  * Reading your feedback here: you are spot on! We concede that our claim is misleading. Computing a single row of $T^k$ via repeated sparse matrix-vector multiplications can be accomplished in $\mathcal{O}(mk)$. Therefore computation of all rows of $T^k$ using this approach takes $\mathcal{O}(nmk)$. Thank you for pointing this out! We have corrected the text in the paper.
>
> ## Typos
>
>  * We fixed all typos. You may verify in the revised version uploaded above. Thank you!

---

### Author Response · Authors · 2020-11-25
**Refactored code & producing large experiments table**

# Implementation Update

**Note: If you care about the practical aspect of our contribution, this can be relevant to you but if you only care about the algorithmic and mathematical angle, you may skim / skip this update**

We have spent efforts over the last 2 weeks to rewrite our code. Now it is extremely clean and modular. More importantly, it has a few key performance enhancement including:
* Utilization of data loaders of pytorch and tensorflow. Now the Walk Forests are generated in parallel, as models are computing.
* Efficient "trimming" and "untrimming" (this is our terminology). Many message-passing models compute $AXW$, where $A$ is some form of normalized adjacency, $X$ is node feature matrix (can also be node latents) and $W$ is trainable weight matrix. Starting from seed (batch) nodes, our walk forest traversal outputs extremely sparse $A$. While multiplication $AX$ is very efficient (using TF/Torch sparse-times-dense), it results in many rows of $X$ being zero (where traversal did not reach) and therefore some of the computation of $(AX)W$ is wasted as $W$ will be multiplied by many zero-rows producing zero-rows, unless the practitioner is careful. To resolve this, we offer this trimming and untrimming natively. With this, we can train small datasets in sometimes less than a second or two, and it has allowed us to run some complex models (e.g. 60-layer GCNII) over large datasets (Redit) in ~2 minutes per epoch.
* We were slightly able to improve some metrics. E.g. we found hyperparameters for GCNII that make us outperform the authors' published results (we just their deep model thinner for pubmed, but same depth) [this also shows improvement on "validation" partition i.e. reflects test accuracy with proper model selection]
* At this point, we have made many shell scripts as `<method>_on_<dataset>.sh`   and each invoke a runner (there is one runner for pytorch, and another for tensorflow)

We are now compiling a comprehensive table of results that we might add to the main paper or the appendix, since we get a 9th page with your recommendation for acceptance. We summarize it here (many entries are running or scheduled to run).

|  | Cora | Citeseer | Pubmed | Redit |
|:---:|:---:|:---:|:---:|:---:|
| GAT |     83.2         |     72.4  | 77.7          |     OOM       |
| F(GAT) |  83.3  |   72.5        |    77.8           |     ?       |
| GCN |     81.5     | 70.3     |      79.0         |       OOM     |
| F(GCN) | 81.9     |  69.8      |    79.4        |        94.9    |
| MixHop |  81.9     |  71.4       |    80.8           |        OOM    |
| F(MixHop)| 83.1    |  71.8         |   80.9           |       94.5     |
| SimpleGCN |  81.0 |     71.9      |       78.9        |      94.9 on CPU OOM on GPU    |
| F(SimpleGCN) |  81.0  |  71.9         |   79.0            |      94.8      |
| GCNII |    85.5  |   73.4     |     80.3          |     OOM      |
| F(GCNII) | 85.9   |     73.4     |      80.7      |     ? 94.1       |

Notes:

* ? means scheduled or currently running (i.e. results may improve)
* SimpleGCN has a convex objective (only one trainable layer). Our runs have no variance (even though initializations are random). We probably converge to a slightly different point that SimpleGCN, perhaps because we may not match exactly the weight decay (we use TF and they use Torch)
* Some numbers are copied from other papers and from our previous experiments. We updated some of our numbers using the freshest code. We will update all numbers before camera ready, if paper gets accepted, to match output of the shell scripts.
* More rows and columns will be added

---

### Decision · Program_Chairs · 2021-01-07
**Final Decision**

**Decision:**

Accept (Poster)

**Comment:**

Summary: The authors propose to approximate operations on graphs, roughly
speaking by approximating the graph locally around a collection of
vertices by a collection of trees. The method is presented as a
meta-algorithm that can be applied to a range of problems in the
context of learning graph representations.

Discussion: The reviews are overall positive, though they point out a
number of weaknesses. One was unconvincing experimental
validation. Another, more conceptual one was that this is a 'unifying
framework' rather than a novel method. Additionally, there were a number of
minor points that were not clear. However, the authors have provided
additional experiments that the reviewers consider convincing, and
were able to provide sufficient clarification.

Recommendation:
The reviewer's verdict post-discussion favors publication, and I
agree. The authors have convincingly addressed the main concerns in discussion, and novelty is not a necessity: Unifying frameworks often seem an end in themselves, but this one is
potentially useful and compellingly simple.